# Transcriptional Activation of Ecdysone-Responsive Genes Requires H3K27 Acetylation at Enhancers

**DOI:** 10.3390/ijms231810791

**Published:** 2022-09-16

**Authors:** Dong Cheng, Zhaoming Dong, Ping Lin, Guanwang Shen, Qingyou Xia

**Affiliations:** 1State Key Laboratory of Silkworm Genome Biology, Biological Science Research Center, Southwest University, Chongqing 400715, China; 2Chongqing Key Laboratory of Sericultural Science, Southwest University, Chongqing 400715, China

**Keywords:** chromatin regulation, cisregulatory elements, ecdysone receptor, gene transcription, histone modification

## Abstract

The steroid hormone ecdysone regulates insect development via its nuclear receptor (the EcR protein), which functions as a ligand-dependent transcription factor. The EcR regulates target gene expression by binding to ecdysone response elements (EcREs) in their promoter or enhancer regions. Its role in epigenetic regulation and, particularly, in histone acetylation remains to be clarified. Here, we analyzed the dynamics of histone acetylation and demonstrated that the acetylation of histone H3 on lysine 27 (H3K27) at enhancers was required for the transcriptional activation of ecdysone-responsive genes. Western blotting and ChIP-qPCR revealed that ecdysone altered the acetylation of H3K27. For *E75B* and *Hr4*, ecdysone-responsive genes, enhancer activity, and transcription required the histone acetyltransferase activity of the CBP. EcR binding was critical in inducing enhancer activity and H3K27 acetylation. The CREB-binding protein (CBP) HAT domain catalyzed H3K27 acetylation and CBP coactivation with EcR, independent of the presence of ecdysone. Increased H3K27 acetylation promoted chromatin accessibility, with the EcR and CBP mediating a local chromatin opening in response to ecdysone. Hence, epigenetic mechanisms, including the modification of acetylation and chromatin accessibility, controlled ecdysone-dependent gene transcription.

## 1. Introduction

In insects, the steroid hormone 20-hydroxyecdysone (20E) functions as a generalized systemic signaling molecule that coordinates the critical developmental events of embryogenesis, larval molting, metamorphosis, and reproduction [1,2]. The steroid hormone 20E mediates biological activity by binding to a nuclear hormone receptor heterodimer comprising ecdysone receptor (EcR) and ultraspiracle (USP) proteins—orthologs of the vertebrate LXR and RXR receptors, respectively [3]. Similar to other hormone receptors, EcR regulates transcription by binding to specific target-gene deoxyribonucleic acid (DNA) sequences known as ecdysone response elements (EcREs) [4,5,6]. These response elements exist not only in the promoter sequence of the target gene, but also in the enhancer regions, several kilobases upstream of the transcription initiation site [7,8]. Ligand-bound receptors recruit coactivators to modify the chromatin architecture and to allow the subsequent recruitment of additional transcription factors and members of the basal transcriptional machinery for transcription activation [9].

Hormone-triggered transcriptional responses are highly cell-specific, and the chromatin environment surrounding regulatory regions is an important determinant of target-gene transcriptional responses [10,11,12]. The chromatin structure is dynamic, undergoing reversible chemical changes mainly in the form of histone post-translational modifications. Histone acetylation, extensively studied in the context of gene regulation, is highly dynamic and occurs on lysine residues mainly within histone protein N-terminal tail domains [13]. Increased histone residue acetylation is thought to weaken the interaction with DNA, thus, improving accessibility to DNA regulatory elements. Histone deacetylation, in contrast, transcriptionally represses gene expression [14]. Histone acetylation is highly dynamic during insect developmental transitions, which is crucial for gene activation [15,16]. For example, histone acetylation is important in the transcription of Halloween genes, and in juvenile hormone activity [17,18]. In *Drosophila melanogaster* (*D. melanogaster*; Diptera: Drosophilidae), the ecdysone-induced expression of *Eip74EF* and *E75B* was associated with the acetylation of histone H3 on lysine 23 [19]. Although histone acetylation has been widely studied in insects, little is known about its role and activity during ecdysone-induced dynamic transcriptional changes. Thus, the mechanisms underlying ecdysone-induced histone acetylation and changes in gene expression remain to be elucidated.

Histone acetylation is catalyzed by histone acetyltransferase (HAT), and deacetylation is influenced by histone deacetylase (HDAC) [20]. The dynamic interplay between these opposing activities is thought to regulate cellular histone acetylation levels as well as at local promoter and gene levels [21]. As a major histone acetyltransferase, the CREB-binding protein (CBP) possesses intrinsic acetyltransferase activity and acetylates lysine residues to alter chromatin structure, making genes more accessible to transcription [22]. The CBP, which has diverse and important functions in postembryonic insect development and metamorphosis [23,24,25,26], has also been identified in mammals as a transcription coactivator associated with many different transcription factors [27,28]. Nonetheless, little is known about its activity and role during ecdysone-induced dynamic transcriptional changes. A study showed that CBP is crucial for acetylating the histone of *Sox14*, an ecdysone-induced gene crucial for regulating dendrite pruning during *Drosophila* metamorphosis [29]. The CBP forms a protein complex with EcR in an ecdysone-dependent manner, thereby activating *sox14* transcription. Current research on this topic is based mainly on *D. melanogaster*. These studies have mainly focused on gene promoters; few have addressed regulatory regions such as enhancers, or the whole genome [19,30,31]. There is, therefore, a need to examine whether the CBP can associate with EcR as a coactivator to regulate ecdysone-induced gene expression in other insects.

Using ChIP-seq experiments in silkworm BmE cells, it has previously been shown that ecdysone triggers genomic changes in histone modification, and that histone acetylation is important in regulating ecdysone-responsive genes [32]. To better understand the role of histone acetylation in ecdysone-induced gene transcription, we examined the acetylation of histone H3 on lysine 27 (H3K27), in the ecdysone response genes *E75B* and *Hr4*, after treatment with 20E. This treatment rapidly and significantly increased H3K27 acetylation in the regulatory elements of *E75B* and *Hr4* loci. This increase was necessary to activate the regulatory elements. Our results showed that the EcR plays a central role in H3K27 acetylation and chromatin accessibility by recruiting the histone acetyltransferase CBP at enhancers. Thus, we demonstrated that epigenetic regulation plays critical roles in controlling the expression of ecdysone-responsive genes.

## 2. Results

### 2.1. Ecdysone Induces Local Dynamic Changes in H3K27 Acetylation

We classified ecdysone-responsive genes in BmE cells as up- or downregulated using published RNA-seq and ChIP-seq data, and analyzed changes in H3K27 acetylation in these differentially expressed genes. The results showed that the 20E treatment significantly increased H3K27 acetylation in upregulated genes, and reduced it in downregulated genes (Figure 1A). This result revealed H3K27 acetylation as a key epigenetic marker positively associated with ecdysone-responsive gene expression.

The total histone H3K27 acetylation was only affected a little by the 20E treatment at various concentrations (Figure 1B), which suggested that whole-genome H3K27 acetylation did not change significantly in response to 20E. Then, we examined H3K27 acetylation levels at individual gene loci. *E75B*, an early response gene, and *Hr4*, an early–late gene, play key roles in ecdysone-triggered genetic cascades [33,34]. The 20E treatment substantially altered H3K27 acetylation levels in multiple regions of the *E75B* and *Hr4* loci (Figure 1C,E), mainly in noncoding regions such as the promoters, introns, and intergenic regions. We observed similar results for other ecdysone-responsive genes, such as *Hr3*, *E93*, and *Vrille* (Appendix A).

We performed chromatin immunoprecipitation coupled with quantitative PCR (ChIP-qPCR) to examine H3K27 acetylation levels in BmE cells. Consistent with the ChIP-seq results, H3K27 acetylation levels were significantly higher in regions a, b, c, and d of *E75B* and *Hr4*, but lower in regions e and f of *E75B* (Figure 1D,F).

### 2.2. Enhancer H3K27 Acetylation Enables Ecdysone-Responsive Gene Transcription Activation

We cloned these differential H3K27 acetylation regions in *E75B* and *Hr4* loci and tested their activity using a dual luciferase reporter assay; these regions significantly increased reporter gene expression, indicating their function as enhancers (Figure 2A,B). More importantly, the 20E treatment significantly upregulated the activity of the a, b, c, and d enhancer regions of *E75B* and *Hr4* loci, while reducing that of the e and f enhancer regions of *E75B*. These dual luciferase reporter assay-based results were consistent with the H3K27 acetylation dynamics in the enhancer regions of *E75B* and *Hr4* loci derived via ChIP-qPCR (Figure 1D,F). This consistency indicated that H3K27 acetylation reflected the enhancer activity in response to 20E. Further, *E75B* and *Hr4* mRNA expression was significantly upregulated after the 20E treatment (Figure 2C,D). Together, these results indicated that ecdysone induced enhancer activity by altering H3K27 acetylation dynamics, thereby regulating *E75B* and *Hr4* expression.

### 2.3. Enhancer Classification Based on H3K27 Acetylation Dynamics in Response to 20E

We performed the genome-wide quantification of H3K27 acetylation in the identified enhancers [32]. Throughout the genome, the H3K27 acetylation of enhancers changed substantially in response to the 20E treatment (Figure 3A,B, Appendix A). A relatively small subset of enhancers (*n* = 997, ~13.6%) exhibited at least a 1.5-fold increase in H3K27ac in response to 20E. Some enhancers exhibited a high level of H3K27 acetylation before and after the 20E stimulation. We also identified enhancers (*n* = 600, ~8.2%) that underwent a decrease in H3K27 acetylation in response to 20E. We then classified these responses as “increasing”, “constant”, or “decreasing” (Figure 3A,B).

### 2.4. EcR Binding Is Highly Enriched in H3K27 Acetylation-Enriched Enhancers

Strikingly, the de novo motif analysis showed that the EcR motif was the most significantly enriched motif in H3K27 acetylation-increasing enhancers, whereas it was not enriched in the other two categories (Figure 4A). This suggested that the EcR played a key role in promoting the 20E-induced increase in H3K27 acetylation in the enhancers. Other motifs, such as Br-C and Deaf1, were also enriched in H3K27 acetylation-increasing enhancers (Appendix A). The Eip74 motif was most enriched in H3K27 acetylation-decreasing enhancers (Figure 4A). The transcription factor E74, which is induced by ecdysone early in metamorphosis, is a potent repressor of late secondary-response gene transcription [35]. This suggests that E74 may participate in suppressing gene expression by reducing H3K27 acetylation on enhancers. Other motifs, such as Ftz-F1and ERR, were also enriched in H3K27 acetylation-decreasing enhancers (Appendix A).

We selected two H3K27 acetylation-increasing enhancers that drove the expression of a luciferase reporter only upon the use of the 20E treatment, yet responded with different strengths. We introduced point mutations to disrupt the EcR motif (Appendix A); this severely reduced or abolished the inducibility of these two enhancers (Figure 4B). Similarly, to test whether the Eip74 motif was required for the H3K27 acetylation-decreasing enhancer function, we introduced point mutations to repress the enhancers of *Cyp18a1* and *PstC*, which indeed abolished their repression (Figure 4B).

Western blotting revealed that the total H3K27 acetylation was reduced after double-stranded RNA (dsRNA)-induced EcR knockdown in ecdysone-treated BmE cells (Figure 4C). Likewise, ChIP-qPCR revealed a significant reduction in H3K27 acetylation in the enhancer regions of *E75B* and *Hr4* loci (Figure 4D). Hence, the EcR was thought to participate in inducing H3K27 acetylation.

### 2.5. The HAT Domain of CBP Is Required to Acetylate H3K27

The silkworm CBP contains a typical CBP/p300-type histone acetyltransferase domain (Figure 5A), similar in sequence and structure to that of the CBP in *D. melanogaster*, and to the CBP orthologs in other insect species [36,37]. Western blotting revealed that H3K27 acetylation was largely abolished after the CBP knockdown in the absence of HDAC inhibition, whereas levels remained high in the EGFP control (Figure 4C). We further examined the changes in H3K27 acetylation in the enhancer regions of *E75B* and *Hr4* loci after the CBP knockdown via ChIP-qPCR; these levels were substantially lower after the CBP knockdown (Figure 5B). Consistent with this, the CBP knockdown in BmE cells also reduced *E75B* and *Hr4* mRNA expression (Figure 5C). These results suggest that, in BmE cells, CBP predominantly mediates H3K27ac levels and its action is, thus, necessary for ecdysone-responsive gene expression.

We treated cells with two CBP inhibitors, C646, which specifically inhibited its HAT domain activity, and SGC-CBP30, which specifically inhibited its bromodomain activity. A brief (1 h) treatment with C646, which was sufficient to suppress H3K27 acetylation (Figure 5D), fully prevented *E75B* and *Hr4* mRNA expression (Figure 5E). This suggested that the CBP HAT domain was necessary to induce H3K27 acetylation. Although SGC-CBP30 also inhibited *E75B* and *Hr4* mRNA expression, it did not affect H3K27 acetylation under the same conditions (Figure 5D). These results suggested that 20E-induced H3K27 acetylation was independent of the CBP bromodomain, although the inhibitory effects on mRNA induction implied that the domain may have participated in 20E-regulated transcription. In addition, we treated cells with the broad-spectrum HDAC deacetylase family inhibitor trichostatin A (TSA) for 12 h; this significantly facilitated both mRNA upregulation and an increase in H3K27 acetylation (Figure 5D,E). These results suggested that the CBP, via its HAT activity, facilitated H3K27 acetylation in enhancer regions and activated ecdysone-responsive gene expression.

### 2.6. The Ecdysone Receptor Recruits CBP as a Coactivator to Acetylate H3K27 in Response to 20E

We found that *CBP* mRNA expression remained unaffected after the EcR knockdown (Figure 6A), indicating that the CBP expression was independent of ecdysone signaling. The coimmunoprecipitation (co-IP) experiments showed that the EcR was detected in the immune complex, when the CBP was immunoprecipitated using an anti-HA antibody (Figure 6B). Interestingly, this interaction was not significantly influenced by 20E, suggesting that the CBP and EcR complex formation was independent of 20E.

The ChIP-qPCR results showed that EcR was enriched in the enhancer regions of *E75B* and *Hr4*, and the enrichment increased after the 20E treatment, indicating that the EcR bound to the enhancer regions to regulate gene expression (Figure 6C). The enrichment of the CBP was very low before the 20E treatment, but was significantly increased after the 20E treatment, indicating that the CBP was recruited to the *E75B* and *Hr4* enhancer regions in response to 20E. Furthermore, we compared the enhancer sequences with the consensus EcRE of silkworms, and found that the enhancer regions of *E75B* and *Hr4* loci indeed contained EcREs, indicating that the EcR could directly bind to these enhancer regions (Appendix A). Thus, our results indicated that the EcR and CBP complexes bound to the enhancer regions after the 20E treatment, and together promoted local acetylation to activate *E75B* and *Hr4* expression.

### 2.7. Ecdysone-Induced Chromatin Opening

Dynamic changes in the chromatin accessibility of enhancers or promoters are important in regulating gene expression. Histone acetylation loosens closed chromatin and makes DNA accessible, facilitating transcription-factor binding to DNA. The formaldehyde-assisted isolation of regulatory element assays coupled with the deep sequencing (FAIRE-seq) signal changed dynamically across the genome following the 20E treatment. For example, at the and *E75B* and *Hr4* loci, the FAIRE-seq signal increased or decreased in multiple enhancer regions upon the use of the 20E treatment, consistent with the changes in H3K27 acetylation (Appendix A). We then quantified the FAIRE-seq signals for our three enhancer categories; after the 20E treatment, 55% of the H3K27 acetylation-increasing enhancers showed an increased FAIRE-signal, whereas for the majority of the H3K27 acetylation-decreased and -constant enhancers, accessibility was on average unchanged (Figure 7A). This suggests that the 20E-induced H3K27 acetylation on enhancers promotes chromatin opening.

The FAIRE-qPCR results showed that the chromatin accessibility of *E75B* and *Hr4* loci increased significantly after the 20E treatment (Figure 7B), consistent with the increase in H3K27 acetylation on the enhancers. ChIP-qPCR revealed that H3K27me3 levels in the enhancer regions of *E75B* and *Hr4* loci were significantly reduced after the 20E treatment (Appendix A). H3K27me3 is believed to participate in silencing gene expression, and is usually associated with heterochromatin [38].

Furthermore, the FAIRE-qPCR results showed that chromatin accessibility was significantly reduced after the EcR or CBP knockdown, even after the 20E treatment (Figure 7C). This shows that ecdysone signaling functionally targets predominantly closed chromatin, suggesting a model in which the EcR, potentially with the coactivator CBP, mediates chromatin opening.

## 3. Discussion

We aimed to clarify the role of ecdysone in epigenetic regulation and, particularly, in histone acetylation. Ecdysone triggered *E75B* and *Hr4* expression by altering histone acetylation and chromatin accessibility, suggesting that H3K27 acetylation and chromatin accessibility to regulatory elements are necessary for activating ecdysone-responsive gene expression. Ecdysone receptors played a key role in acetylating H3K27 by forming a protein complex with the coactivator CBP in a 20E-independent manner. The CBP HAT activity catalyzed histone acetylation, thereby neutralizing lysine residues, in turn increasing regulatory element accessibility to cellular transcriptional machinery and increasing gene expression.

H3K27 acetylation is associated with increased transcriptional activation. Following the 20E treatment in BmE cells, H3K27 acetylation was revealed as a highly dynamic histone marker, positively correlated with gene expression. The activation of ecdysone-responsive gene regulatory elements requires an increase in H3K27 acetylation, which occurs primarily in the proximal and distal regulatory regions of these genes. H3K27 acetylation is well-known as a marker for active enhancers. H3K27 acetylation distinguishes active enhancers from inactive (poised) enhancers containing H3K4me1 alone [39]. In the absence of ecdysone, ecdysone-responsive gene enhancers are often inactive, or have very low activity. Here, the ecdysone treatment caused a rapid increase in H3K27 acetylation at the activated enhancers. Changes in enhancer H3K27 acetylation, therefore, reflected enhancer responses to 20E. Genes typically contain multiple enhancers, which are activated differently depending on the conditions, thus, determining the specificity of gene expression [40]. Therefore, H2K27 acetylation at regulatory elements contributes to specific gene expression programs that determine the cell state and the potential for differentiation into new cell types.

Based on the H3K27 acetylation signal, we categorized the previously identified enhancers [32]. The de novo motif analysis revealed that the EcR motif was the most enriched in H3K27 acetylation-increasing enhancers, consistent with previous studies, suggesting that these enhancers were directly regulated by the 20E-dependent EcR recruitment to chromatin [41]. Importantly, perturbing the EcR activity abolished the 20E-dependent acetylation, indicating that the EcR is central in inducing H3K27 acetylation at these enhancers. H3K27 acetylation-increasing enhancers, which contained the EcR motif, were also strongly enriched in the motifs of putative partner transcription factors required for enhancer function. Therefore, the combination of the EcR and other factors is an important prerequisite in cell type-specific transcriptional responses [42].

Furthermore, enhancer activity repression after the 20E treatment appeared to be independent of the EcR binding, but seemed to involve transcription factors such as Eip74 and Ftz-F1. Given that the Eip74-motif-mutant enhancer also showed weaker activity in the absence of ecdysone (Appendix A), Eip74 may compete with an activator, which is something widely known in other hormone signaling pathways (e.g., glucocorticoid and estrogen receptors) [43]. Therefore, we concluded that the EcR binding was required for the 20E-dependent function of H3K27 acetylation-increasing enhancers, while Eip74 binding was required for the function of H3K27 acetylation-decreasing enhancers.

The CBP is the primary histone acetyltransferase in insects [36,44,45]. The CBP activated the expression of key ecdysone response genes via its intrinsic HAT activity. A recent study showed that the EcR enhances the expression of *C/EBPg*, which binds to the *CBP* promoter, activates *CBP* expression, and leads to histone H3K27 acetylation [46]. Our results suggested that the *CBP* is a coactivator that forms a protein complex with the EcR in an ecdysone-independent manner. Studies in humans, mice, and other mammals have shown that the CBP and its family member p300 function as key transcriptional coactivators, interacting with hundreds of different factors, including hormonal nuclear receptors, to activate gene expression [47]. The CBP acts as a coactivator and is recruited by nuclear receptors [48]. At its amino-terminus, the CBP has a nuclear hormone receptor-binding domain that is potentially associated with the EcR. Hormone receptors can recruit different types of coregulators to regulate gene expression by modifying, or opening or closing, chromatin [9,48]. In various coregulatory complexes, EcR/USP heterodimers regulate gene expression; these include the chromatin remodeler NURF, Dmi-2, SMRTER, the lysine methyltransferase TRR, and the acetyltransferases p160, SRC, and Taiman [31,49,50,51,52]. Our results, therefore, illustrated that intrinsic epigenetic machinery cooperates with systemic steroid hormones to alter chromatin states, and selectively activates critical downstream transcriptional programs.

The organization of accessible chromatin across the genome reflects a network of possible physical interactions, in which enhancers, promoters, insulators, and chromatin-binding factors cooperatively regulate gene expression. Here, the local chromatin of enhancers opened after the 20E treatment, and chromatin accessibility was consistent with increased H3K27 acetylation. Prior to hormone treatment, chromatin remains closed, preventing access to transcription factors, and thereby preventing abnormal gene activation. Histone acetylation has been proposed to facilitate transcription by directly modulating histone–DNA contacts, or by targeting chromatin remodelers to disrupt nucleosomes [53]. More importantly, the accessibility of chromatin decreased following the EcR knockdown, indicating that the EcR can mediate chromatin opening. A prevailing model presents pioneer factors that facilitate chromatin remodeling, thereby allowing the binding of secondary transcription factors, including hormone receptors, which would otherwise be unable to access their target DNA sequence [54]. However, recent studies have shown that steroid receptors can initiate chromatin opening at some sites, with the parallel recruitment of factors often considered to have unique pioneer functions [55]. Further studies are needed to clarify how steroid hormones mediate chromatin opening and closing.

In conclusion, these findings showed that the EcR plays a central role in H3K27 acetylation. Like other nuclear hormone receptors, ligand binding to the EcR induces structural changes that result in differential association with coregulator complexes. In the presence of 20E, the EcR binding at the target enhancer elements of ecdysone-responsive genes would increase H3K27 acetylation by recruiting the coactivator CBP. The CBP, via its intrinsic HAT activity, directly catalyzes the local H3K27 acetylation of enhancers. Histone acetylation neutralizes the positive charge of the lysine residue and results in a loose structure of chromatin accessible for transcriptional machinery, thereby activating gene expression (Appendix A).

## 4. Materials and Methods

### 4.1. Cell Culture, Ecdysone Incubation, and Inhibitor Treatment

The BmE silkworm embryo cell line was maintained in Grace’s insect cell culture medium (Thermo Fisher, Waltham, MA, USA) supplemented with 10% fetal bovine serum (Gibco, Waltham, MA, USA) at 27 °C. Cells were incubated with 2.5 μM ecdysone (H5142; Sigma-Aldrich, St Louis, MO, USA) or 2 μM trichostatin A (S1045; Selleck Chemicals, Houston, TX, USA) in culture medium for 12 h, unless otherwise stated. BmE cells were treated with 30 μM C646 (S7152; Selleck Chemicals, Houston, TX, USA) or 25 μM SGC-CBP30 (S7256; Selleck Chemicals, Houston, TX, USA) for 30 min before harvesting. As a vehicle control, the cells were treated with DMSO in parallel.

### 4.2. qRT-PCR

Total RNA from BmE cells was extracted using an RNeasy Mini Kit (74104; Qiagen, Hilden, Germany). cDNA was synthesized from 500 ng of total RNA using the GoScript Reverse Transcription System (A5001; Promega, Madison, WI, USA), as per the manufacturer’s instructions. RT-qPCR was performed using TB Green Premix Ex Taq II (RR820A; Takara, Kusatsu, Japan) on a 7500 Fast Real-Time PCR System (Thermo Fisher, Waltham, MA, USA). Amplifications were performed in triplicate. Triplicate mean values were calculated using delta–delta Ct quantification, with *RP49* transcription as a normalization reference. The primers used for RT-qPCR analyses are listed in Appendix A.

### 4.3. Transfection and Luciferase Reporter Assay

BmE cells in the logarithmic growth phase were inoculated in 12- or 24-well culture plates and cultured for 18 h. Cell transfection was conducted when the cells reached approximately 80% density. Regions of ca. 1 kb around the center of regulatory elements were PCR-amplified from silkworm genomic DNA and cloned into the pGL3-derived reporter vector. The primer sequences are provided in Appendix A. All constructs were sequenced via Sanger sequencing and cotransfected with the pRL-VgP78M vector as a transfection control using the X-tremeGENE HP DNA Transfection Reagent (XTGHP-RO; Sigma-Aldrich, St Louis, MO, USA). After transfection for 48 h, the BmE cells were treated with 20E for 12 h. Regulatory element activity was measured via luciferase assay using the Dual-Luciferase Reporter Assay System (E1910; Promega, Madison, WI, USA), according to the manufacturer’s instructions. Three replicates were performed for each candidate enhancer.

### 4.4. RNAi

For BmEcR or BmCBP RNAi, double-stranded RNAs (dsRNAs) were synthesized using the T7 RiboMAX Express RNAi System (P1700; Promega, Madison, WI, USA), following the manufacturer’s instructions, with EGFP dsRNA as the control. dsRNA was transfected or cotransfected with pGL3-derived reporter plasmids to determine gene expression or regulatory element activity. Specific PCR primers for dsRNA synthesis were designed for this study (Appendix A).

### 4.5. Western Blotting

For immunoblotting, protein extracts were prepared by lysing BmE cells in RIPA buffer (20 mM Tris-HCI pH7.5, 150 mM NaCl, 10 mM EDTA, 1% Triton X-100, 0.5% sodium deoxycholate, and 0.1% SDS) and boiling in SDS sample buffer for 5 min. Protein samples were resolved on 12% sodium dodecyl sulfate-polyacrylamide gel electrophoresis gels, transferred onto activated polyvinylidene difluoride membranes, and blocked for 1 h in 5% skim milk. Membranes were incubated with primary antibodies against the EcR (DDA2.7, 1:1000; Developmental Studies Hybridoma Bank (DSHB), created by the NICHD of the NIH, maintained at The University of Iowa, Department of Biology, Iowa City, IA, USA), CBP (aa1269-1580, 1:5000; homemade), FLAG (F7425, 1:5000; Sigma-Aldrich, St Louis, MO, USA), HA (AH158, 1:1000; Beyotime, Shanghai, China), H3K27ac (ab177178, 1:10,000; Abcam, Cambridge, UK), and α-tubulin (AF5012, 1:5000; Beyotime, Shanghai, China) overnight at 4 °C, then further incubated with HRP conjugated secondary antibody for 1 h. Signals were detected using the SuperSignal West Pico Chemiluminescent Substrate (34580; Thermo Fisher, Waltham, MA, USA).

### 4.6. Coimmunoprecipitation (Co-IP)

After the cotransfection of the pSL1180-A4 vectors overexpressing HA-tagged N-terminal CBP (aa1–494) and FLAG-tagged EcR in BmE cells, cells were treated with 2.5 μM 20E or DMSO for 12 h. BmE cells were homogenized with lysis buffer (25 mM Tris pH8, 27.5 mM NaCl, 20 mM KCl, 25 mM sucrose, 10 mM EDTA, 10 mM EGTA, 1 mM DTT, 10% (*v*/*v*) glycerol, and 0.5% NP40) with protease inhibitors (cOmplete Cocktail, Roche, Basel, Switzerland) in the presence or absence of 20E. The supernatants were used for immunoprecipitation with either anti-HA (AH158; Beyotime, Shanghai, China) or anti-IgG (AP160; Merck, Darmstadt, Germany) antibodies overnight at 4 °C, followed by incubation with protein A/G Dynabeads (10003D; Thermo Fischer, Waltham, MA, USA) for 2 h. The protein A/G Dynabeads were washed four times with cold PBS. Bound proteins were separated via SDS-PAGE and analyzed via Western blotting using anti-FLAG, anti-HA, and anti-CBP antibodies.

### 4.7. ChIP-qPCR

Chromatin immunoprecipitation (ChIP) assays were performed as previously described. Briefly, treated BmE cells were fixed with formaldehyde and subjected to overnight immunoprecipitation using the corresponding antibodies. The following antibodies were used: anti-H3K27 acetylation (ab4729; Abcam, Cambridge, UK), anti-H3K27me3 (ab6002; Abcam, Cambridge, UK), anti-EcR (DDA2.7; DSHB, Iowa City, IA, USA), and anti-CBP (aa1269-1580; homemade). Purified DNA fragments from immunoprecipitated products were used for qPCR validation. The primers used to amplify the specific region covering the potential regulatory elements for *E75B* or *Hr4*, and the nonspecific regions, are listed in Appendix A. ChIP assay with a nonspecific rabbit IgG antibody was used as a negative control. Enrichment was presented as the percent input for all tested 20E response genes.

### 4.8. FAIRE-Seq and FAIRE-qPCR

To understand the role of chromatin accessibility in the transcription of ecdysone-responsive genes, we performed a FAIRE-seq in BmE cells after the 20E treatment. FAIRE was performed as previously described, with minor modifications [56]. Briefly, formaldehyde-fixed cells (1 × 10^7^ BmE cells) were lysed, and the chromatin was sonicated to a size range of 300–800 bp using a Covaris S220 focused ultrasonicator (15% duty factor, 500 pulse and 500 burst, 20 cycles; Covaris, Woburn, MA, USA). FAIRE DNA was recovered from the cell lysate via phenol–chloroform extraction, in which open chromatin DNA was recovered from the aqueous phase. The recovered chromatin was treated with RNAse A and proteinase K, then purified using a Zymo spin column (C1003-50; Zymo Research, Irvine, CA, USA). Further, FAIRE DNA was quantified via qPCR, to generate a FAIRE–seq library. The primers used for FAIRE-qPCR were the same as those used for ChIP-qPCR. The FAIRE-seq library was built using a NEBNext Ultra II DNA Library Prep Kit (E7645S; NEB, Ipswich, MA) and sequenced on an Illumina HiSeq 2500 (Illumina, San Diego, CA, USA) to generate 150 bp paired-end reads.

### 4.9. Computational Analysis

We used bowtie2 (version 2.3.4.1) to map the clean paired-end reads to the silkworm reference genome [57], which was downloaded from KAIKObase. Available online: https://kaikobase.dna.affrc.go.jp/ (accessed on 20 August 2021). Only uniquely aligned reads were used. Highly enriched peaks were obtained using MACS2 using standard settings [58], allowing for minor modifications (“—nomodel—shift 50—extent 100–q 0.02”). These peaks were hereafter referred to as FAIRE peaks. WIG files, generated using bedtools (version 2.26.0), were, subsequently, used for visualization purposes and to obtain the average signal profiles.

We identified enhancers with H3K4me1 modifications located >1.5 kb beyond the TSS. The number of input-normalized H3K27 acetylation ChIP-seq reads within a 1.5 kb window centered on each enhancer was taken to be the ChIP-seq signal for the enhancer. The enhancers were categorized based on their H3K27 acetylation signals: “increasing” reflected an increase of ≥1.5-fold in H3K27 the acetylation signal upon the 20E stimulation, with the stimulated H3K27 acetylation signal being above the bottom quartile of the stimulated signals; “decreasing” reflected a reduction of ≥33% in the H3K27 acetylation signal upon the 20E stimulation, with the unstimulated signal being above the bottom quartile of unstimulated signals; “constant” referred to the H3K27 acetylation signals in the top quartile of signals in both the unstimulated and stimulated conditions, and with a signal change of ≤10% upon stimulation; and “no response” referred to the enhancers whose H3K27 acetylation peaks did not overlap with those within the H3K27 acetylation datasets. The FAIRE-seq read density for the increasing, decreasing, and constant categories was calculated as described earlier.

To identify the transcription factors involved in the 20E-induced alteration of H3K27 acetylation in enhancers, we performed a de novo motif analysis using these three categories. De novo motifs were identified using HOMER motif tools (findMotifs.pl) [59]. Motifs were also screened using RSAT peak motifs with the default parameters [60]. We compared the identified motifs with those in the JASPAR core nonredundant insect database (2020), the RSAT nonredundant insect database (2017), and the DrosophilaTFs database (2015).

## Figures and Tables

**Figure 1 ijms-23-10791-f001:**
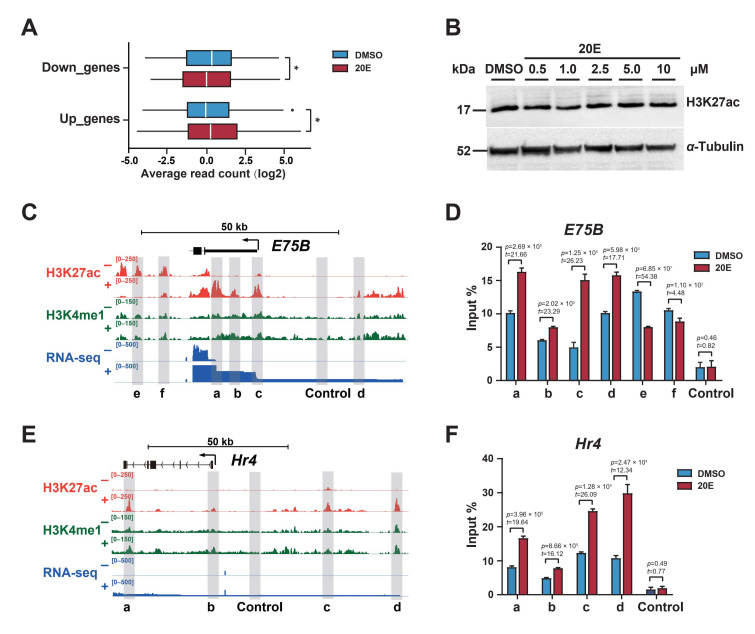
Ecdysone induced dynamic changes in local H3K27 acetylation. (**A**) Changes in H3K27 acetylation ChIP-seq signals for the differentially expressed genes (log_2_ scale). The differentially expressed genes (“Up_genes” and “Down_genes”, respectively) were defined as those with at least a 1.5-fold change in expression after 20E treatment. * *p* < 0.05; Student’s *t*-test. (**B**) Western blotting of H3K27 acetylation levels in dimethyl sulfoxide (DMSO) or 20Etreated BmE cells. The antibodies used are shown on the right. (**C**) and (**E**) show the IGV genome browser screenshots of ChIP-seq and RNA-seq tracks for the *E75B* and *Hr4* gene loci. Grey shading highlights putative enhancer regions. Control regions lacking enhancer chromatin were also chosen from the gene loci. + and − indicate the presence and absence of 20E treatment, respectively. (**D**) and (**F**) show the ChIP-qPCR analysis of H3K27 acetylation in DMSO or 20Etreated BmE cells. DNA was quantified using primers designed to amplify the putative regulatory elements of the *E75B* and *Hr4* gene loci shown in (**B**). Error bars: standard errors of the mean (SEM) from three biological replicates measured in duplicate. Student’s *t*-test (t = 2.78, df = 4).

**Figure 2 ijms-23-10791-f002:**
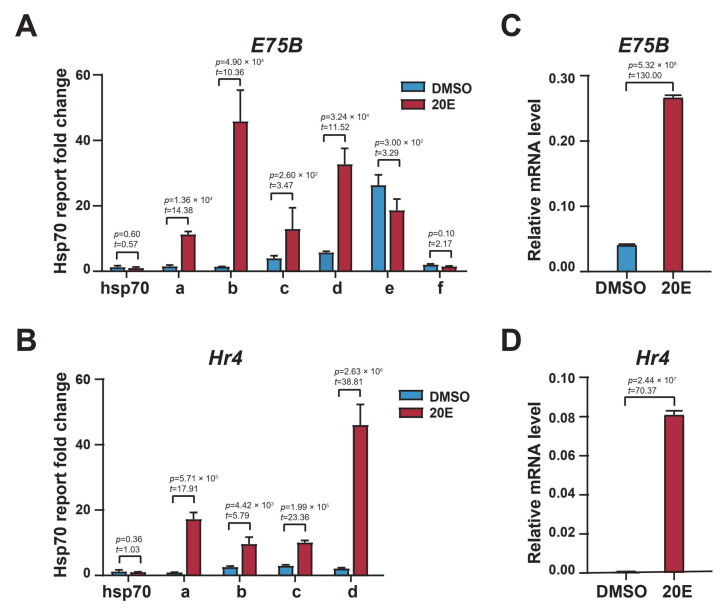
H3K27 acetylation was required for *E75B* and *Hr4* enhancer activity and mRNA expression. (**A**,**B**) Functional testing of enhancers in DMSO or 20Etreated BmE cells via luciferase assays. Enhancers were cloned using primers designed to amplify the putative regulatory elements of *E75B* and *Hr4* loci. The changes shown were fold increases over the pGL3 reporter backbone with the *hsp70* core promoter. (**C**,**D**) RT–qPCR of *E75B* and *Hr4* expression in DMSO- or 20E-treated BmE cells. We calculated the ratios of the RNA levels of the gene of interest to those of *RP49* in DMSO- and 20E-treated cells. Error bars: standard error of the mean (SEM) from three biological replicates measured in duplicate. Student’s *t*-test (df = 4).

**Figure 3 ijms-23-10791-f003:**
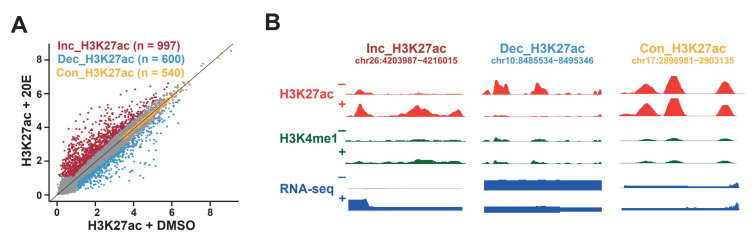
Enhancers were categorized based on changes in H3K27 acetylation. (**A**) Enhancer categorization based on H3K27 acetylation signals in response to 20E treatment. H3K27 acetylation signals were taken from the normalized H3K27 acetylation ChIP-seq reads within a 1.5 kb window centered on each enhancer. Inc, increasing; Con, constant; Dec, decreasing. (**B**) IGV genome browser screenshots of ChIP-seq and RNA-seq tracks for representative loci demonstrating distinct H3K27 acetylation dynamics at enhancers in response to 20E treatment. + and − indicate the presence or absence of 20E treatment, respectively.

**Figure 4 ijms-23-10791-f004:**
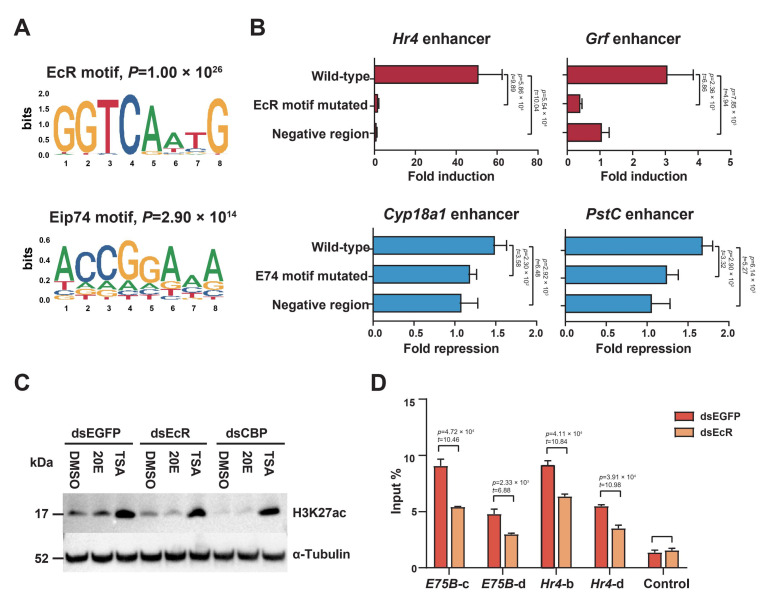
Ecdysone receptor (EcR) played a critical role in inducing enhancer activity. (**A**) Upper and lower panels: de novo motif analysis of enhancers with increasing or decreasing H3K27 acetylation, respectively. The most enriched motif is shown. (**B**) Luciferase assays for the enhancer categories in DMSO- or 20E-treated BmE cells. Fold induction of the normalized luciferase signal for the H3K27 acetylation-increasing enhancers near *Hr4* and *Grf* assessing wild-type sequences and EcR-motif-mutated variants. Fold repression of normalized luciferase signal for H3K27 acetylation-decreasing enhancers near *Cyp18a1* and *PstC* assessing wild-type and Eip74-motif-mutated variants, in which the enhancer did not contain an EcR motif. Negative region: sequence from the *Hr4* control region, as in Figure 1E. The DNA sequences are listed in Appendix A. Student’s *t*-test (df = 4). (**C**) Western blotting of protein extracts of BmE cells first treated with dsRNA against EGFP, EcR, or CBP, then either left untreated or exposed to 20E or TSA for 12 h. The antibodies used are shown on the right. (**D**) ChIP-qPCR of H3K27 acetylation in BmE cells treated with dsRNA against EGFP or EcR. DNA was quantified using primers designed to amplify the enhancer regions of *E75B* and *Hr4* gene loci, as shown in Figure 1C,E. Error bars: standard error of the mean (SEM) from three biological replicates measured in duplicate. Student’s *t*-test (df = 4).

**Figure 5 ijms-23-10791-f005:**
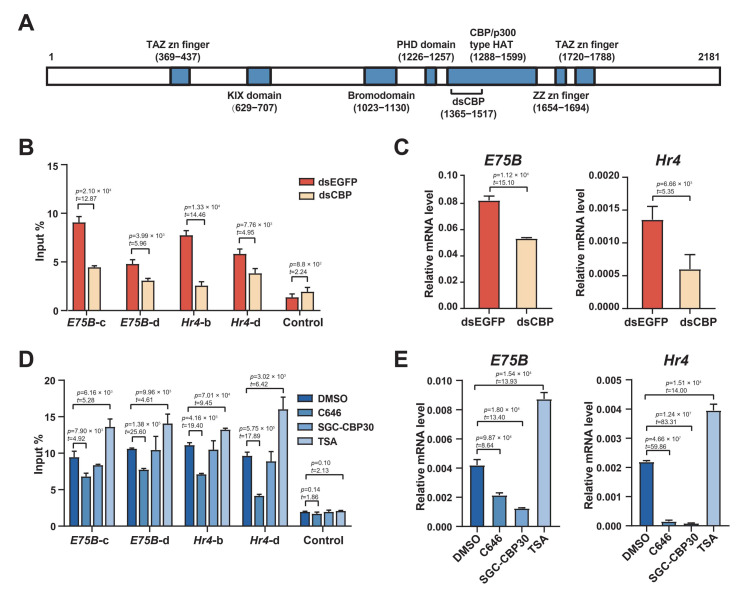
CBP HAT activity was critical for ecdysone-responsive gene expression. (**A**) Scheme of CBP organization, indicating the main functional motifs and the sites targeted by the dsRNA. Length is proportional to the lengths of amino acids. (**B**) ChIP-qPCR of H3K27 acetylation in BmE cells treated with dsRNA against EGFP or CBP. DNA was quantified using primers designed to amplify the enhancer regions of *E75B* and *Hr4* gene loci, as shown in Figure 1C,E. Error bars: standard error of the mean (SEM) from three biological replicates measured in duplicate. Student’s *t*-test (df = 4). (**C**) RT–qPCR analysis of *E75B* and *Hr4* expression in BmE cells treated with dsRNA against EGFP or CBP. Ratios of RNA levels of the gene of interest to *RP49* RNA levels were calculated. Error bars: standard error of the mean (SEM) from three biological replicates measured in duplicate. Student’s *t*-test (df = 4). (**D**) ChIP-qPCR of H3K27 acetylation in BmE cells treated with DMSO, C646 (a CBP HAT domain-specific inhibitor), SGC-CBP30 (a CBP-bromodomain-specific inhibitor), or trichostatin A (TSA, a HDAC inhibitor), respectively. The DNA was quantified using primers designed to amplify the enhancer regions of *E75B* and *Hr4* gene loci, as shown in Figure 1C,E. Error bars: standard error of the mean (SEM) from three biological replicates measured in duplicate. Student’s *t*-test (df = 4). (**E**) RT-qPCR of *E75B* and *Hr4* expression in BmE cells treated with DMSO, C646, SGC-CBP30, and TSA, respectively. We calculated the ratios of the RNA levels of the gene of interest to those of *RP49*. Error bars: standard error of the mean (SEM) from three biological replicates measured in duplicate. Student’s *t*-test (df = 4).

**Figure 6 ijms-23-10791-f006:**
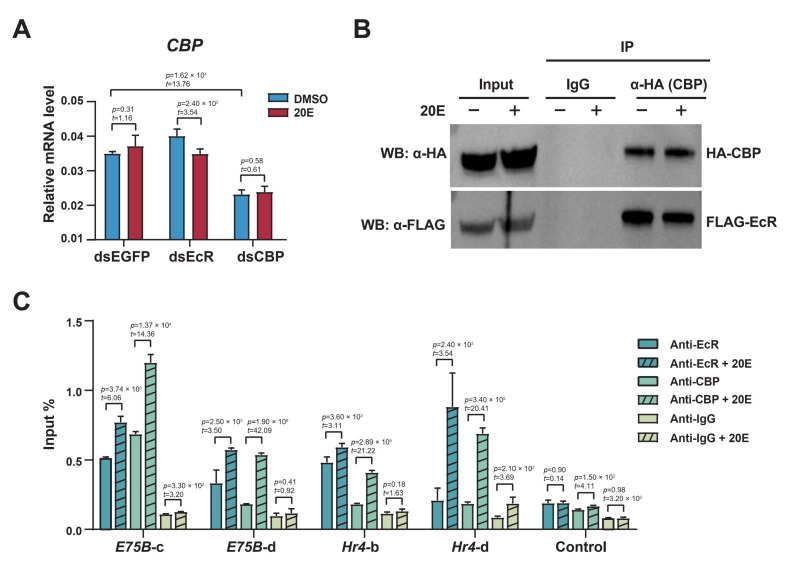
Ecdysone receptor (EcR) cooperated with CBP-mediated histone acetylation in response to 20E. (**A**) RT-qPCR of *CBP* expression in BmE cells first treated with dsRNA against EGFP, EcR, or CBP, then either left untreated or exposed to 20E for 12 h. We calculated the ratios of RNA levels of the gene of interest to those of *RP49*. Error bars: standard error of the mean (SEM) from three biological replicates measured in duplicate. Student’s *t*-test (df = 4). (**B**) CBP interacted with EcR in the presence of 20E. DMSO- or 20E-treated BmE cells were cotransfected with HA-CBP and FLAG-EcR. Extracts were immunoprecipitated with HA (using the HA IP antibody). The immunoprecipitates were analyzed via Western blotting. (**C**) ChIP-qPCR analysis of EcR, CBP, and IgG in DMSO- or 20E-treated BmE cells. DNA was quantified using primers designed to amplify the putative regulatory elements of *E75B* and *Hr4* gene loci, as shown in Figure 1C,E. Error bars: standard error of the mean (SEM) from three biological replicates measured in duplicate. Student’s *t*-test (df = 4).

**Figure 7 ijms-23-10791-f007:**
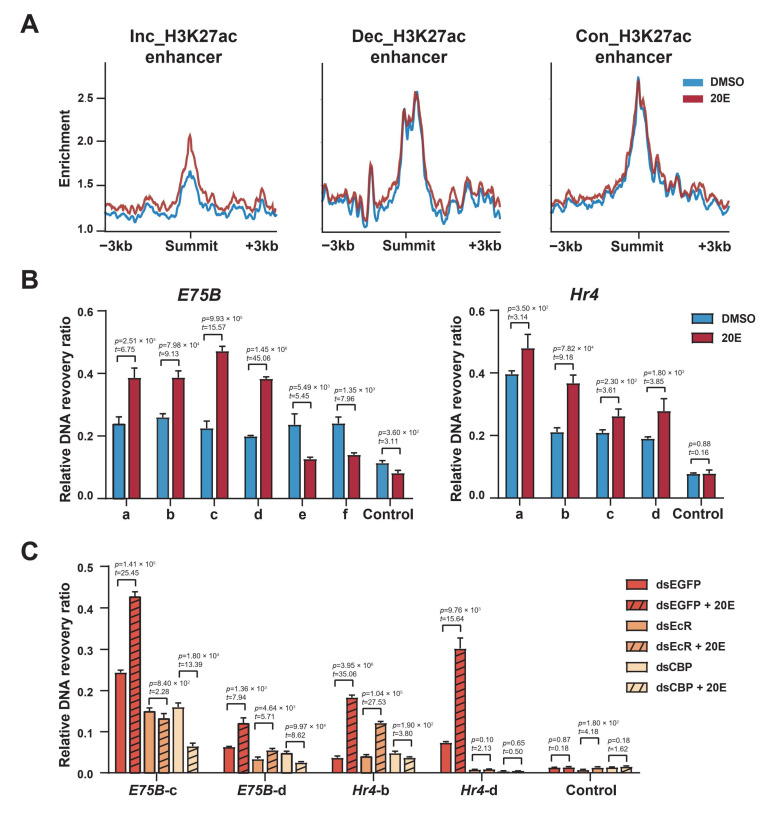
Ecdysone signaling can target inaccessible chromatin. (**A**) FAIRE-seq read density profiles ±3 kb around the summits of enhancers from the three enhancer categories before and after 20E treatment (blue and orange, respectively). Inc_H3K27ac, Dec_H3K27ac, and Con_H3K27ac: increasing, decreasing, and constant H3K27 acetylation, respectively. (**B**) Nucleosome density assayed via FAIRE-qPCR in BmE cells treated with DMSO or 20E. The relative DNA recovery ratio between free versus total DNA was calculated and plotted. DNA was quantified using primers designed to amplify the putative regulatory elements of *E75B* and *Hr4* gene loci. Error bars: standard errors of the mean (SEM) from three biological replicates measured in duplicate. Student’s *t*-test (df = 4). (**C**) Nucleosome density assayed via FAIRE-qPCR in BmE cells first treated with dsRNA against EGFP, EcR, or CBP, then either left untreated or exposed to 20E for 12 h. The relative DNA recovery ratio between free versus total DNA was calculated and plotted. DNA was quantified using primers designed to amplify the putative regulatory elements of *E75B* and *Hr4* gene loci, as shown in Figure 1C,E. Error bars: standard error of the mean (SEM) from three biological replicates measured in duplicate. Student’s *t*-test (df = 4).

## Data Availability

The FAIRE-seq data generated in this study were deposited in the Sequence Read Archive (SRA) under accession numbers SRR19135339 and SRR19135340. Additional data used in this manuscript were deposited in BioProject accession numbers PRJNA450142 (ChIP-seq of H3K27ac) and PRJNA449979 (RNA-seq).

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
