# Peer review of "Transcriptional Activation of Ecdysone-Responsive Genes Requires H3K27 Acetylation at Enhancers"

_ijms, 2022, doi:10.3390/ijms231810791_

Round 1

Reviewer 1 Report

The steroid hormone ecdysone regulates insect development via its nuclear receptor, which functions as a ligand-dependent transcription factor. The paper studies that epigenetic mechanisms including modification of acetylation and chromatin accessibility, control ecdysone-dependent gene transcription. The experiment is well designed and innovative in this field. The introduction provide sufficient background and include all relevant references. The methods are adequately described. 

Author Response

Reviewer #1:

The steroid hormone ecdysone regulates insect development via its nuclear receptor, which functions as a ligand-dependent transcription factor. The paper studies that epigenetic mechanisms including modification of acetylation and chromatin accessibility, control ecdysone-dependent gene transcription. The experiment is well designed and innovative in this field. The introduction provides sufficient background and include all relevant references. The methods are adequately described.

Response: Thanks a lot for your careful reading our work and overall positive evaluation.

Reviewer 2 Report

These are my main comments on the manuscript (ijms-1879961) entitled “Transcriptional activation of ecdysone-responsive genes requires H3K27 acetylation at enhancers”. The manuscript shows that ecdysone altered the acetylation of histone H3 on lysine 27 (H3K27). In general, the findings show that, in the presence of 20E, EcR recruits the coactivator CBP to form a complex. Following substantial revisions should be incorporated in the manuscript prior to acceptance.

A few points:

Ls.9-20: In abstract, provide a main objective to clarify this work.

L.48: For this insect species, provide the ID author scientific name, order and family taxa.

L.68: Drosophila melanogaster should be D. melanogaster.

Ls.72-84: A hypothesis for this work is needed. Also, the authors provide various objectives. Summarize as one main objective.

Ls.85-332: For each Student's t-test, provide the t-value, degree freedom, and estimated p-value. In results section, insert this information in all data analyzed.

Ls.404-408: Any conclusion? You have several results obtained in your investigation but you do not conclude anything. Please expand the conclusion section. Explain in more detail which ecdysone altered the acetylation of histone H3 on lysine 27 (H3K27).

Author Response

Reviewer #2:

These are my main comments on the manuscript (ijms-1879961) entitled “Transcriptional activation of ecdysone-responsive genes requires H3K27 acetylation at enhancers”. The manuscript shows that ecdysone altered the acetylation of histone H3 on lysine 27 (H3K27). In general, the findings show that, in the presence of 20E, EcR recruits the coactivator CBP to form a complex. Following substantial revisions should be incorporated in the manuscript prior to acceptance.

Response: Thank you very much for your professional review, kind suggestions and providing the correct description. We have carefully improved the manuscript according to your comments. The revised portions of the manuscript text are marked in red. Our specific responses to the comments are listed as follows:

A few points:

Ls.9-20: In abstract, provide a main objective to clarify this work.

Response: We thank the reviewer for the valuable suggestion. According to your suggestion, we have added a main objective to clarify this work. The sentence added to the abstract as “Here, we analyzed the dynamics of histone acetylation and demonstrated that the acetylation of histone H3 on lysine 27 (H3K27) at enhancers was required for transcriptional activation of ecdysone-responsive genes.” (Lines 12-15).

L.48: For this insect species, provide the ID author scientific name, order and family taxa.

Response: Thanks for your suggestion. We have provided the scientific name, order and family taxa of this insect species (Drosophila melanogaster (Diptera: Drosophilidae)). (Lines 51-52).

L.68: Drosophila melanogaster should be D. melanogaster.

Response: According to your suggestion, we have replaced “Drosophila melanogaster” with “D. melanogaster”. (Line 72).

Ls.72-84: A hypothesis for this work is needed. Also, the authors provide various objectives. Summarize as one main objective.

Response: Thanks for your suggestion. We have revised the last paragraph of introduction by summarizing these objectives to one main objective and adding a hypothesis for this work. The revised sentence as “Our results showed that EcR plays a central role in H3K27 acetylation and chromatin accessibility, by recruiting the histone acetyltransferase CBP at enhancers. Thus, we demonstrated that epigenetic modification plays critical roles in controlling expression of ecdysone-responsive genes.” (Lines 82-86).

Ls.85-332: For each Student's t-test, provide the t-value, degree freedom, and estimated p-value. In results section, insert this information in all data analyzed.

Response: According to your suggestion, we have provided the statistical analysis (t-value, degree freedom, and estimated p-value) in the new version of manuscript. We inserted this statistical information into the results section.

Ls.404-408: Any conclusion? You have several results obtained in your investigation but you do not conclude anything. Please expand the conclusion section. Explain in more detail which ecdysone altered the acetylation of histone H3 on lysine 27 (H3K27).

Response: According to your suggestion, we have revised and expanded the conclusion: “In conclusion, these findings show that EcR plays a central role in H3K27 acetyla-tion. Like other nuclear hormone receptors, ligand binding to EcR induces structural changes that result in differential association with coregulator complexes. In the presence of 20E, EcR binding at target enhancer elements of ecdysone-responsive genes would increase H3K27 acetylation, by recruiting coactivator CBP. CBP, via its intrinsic HAT activity, directly catalyzes local H3K27 acetylation of enhancers. Histone acetylation neutralizes the positive charge of the lysine residue and results in a loose structure of the chromatin accessible for the transcriptional machinery, thereby activating gene expression.” (Lines 382-390).

Reviewer 3 Report

This manuscript investigates “Transcriptional activation of ecdysone-responsive genes requires H3K27 acetylation at enhancers". This information is of interest to biologists working within insect physiology and merits publication. The experimental set up of this study appears to be well-designed and the data collected carefully. I think that this manuscript requires moderate rewriting to make its results clearer and more readily interpretable to the reader. My specific comments are listed in the "Main text". The authors provide knowledge in this manuscript and can be of great interest to the journal. Based on the comments above reported, my opinion is that this manuscript may be suitable for printing on this journal.

1. I have concerns about the manuscript sections that I believe need to be addressed in order to improve its clarity.

2. Principal main for this research should be clarified.

3. In results section, several aspects (or sentence) should be incorporated in methods and discussion section

4. Other revisions could be checked in PDF attached.

Author Response

Reviewer #3:

This manuscript investigates “Transcriptional activation of ecdysone-responsive genes requires H3K27 acetylation at enhancers". This information is of interest to biologists working within insect physiology and merits publication. The experimental set up of this study appears to be well-designed and the data collected carefully. I think that this manuscript requires moderate rewriting to make its results clearer and more readily interpretable to the reader. My specific comments are listed in the "Main text". The authors provide knowledge in this manuscript and can be of great interest to the journal. Based on the comments above reported, my opinion is that this manuscript may be suitable for printing on this journal.

Response: We would like to express our great appreciation to you for the comments and suggestions regarding our study. The comments are valuable and have been very helpful for revising and improving our manuscript. We have examined the comments carefully and have made corrections to the manuscript accordingly, and here we did not list the changes but marked in green in revised paper.

  1. I have concerns about the manuscript sections that I believe need to be addressed in order to improve its clarity.

Response: Thanks for your suggestion. I have changed the sentence in the revised manuscript according to your suggestion.

  1. Principal main for this research should be clarified.

Response: Thanks for your suggestion about. I have corrected in the revised manuscript. As mentioned in the Reviewer #2, we have added a main objective in the abstract to clarify this work: “Here, we analyzed the dynamics of histone acetylation and demonstrated that the acetylation of histone H3 on lysine 27 (H3K27) at enhancers was required for transcriptional activation of ecdysone-responsive genes.” (Lines 12-15).

Besides, we have revised the last paragraph of introduction by summarizing these objectives to one main objective: “Our results showed that EcR plays a central role in H3K27 acetylation and chromatin accessibility, by recruiting the histone acetyltransferase CBP at enhancers. Thus, we demonstrated that epigenetic modification plays critical roles in controlling expression of ecdysone-responsive genes.” (Lines: 82-86).

  1. In results section, several aspects (or sentence) should be incorporated in methods and discussion section.

Response: Thanks for your suggestion. According to your suggestion, we have revised our presentation and incorporated these aspects into methods and discussion section in the new version of the manuscript (marked in green in revised paper).

  1. Other revisions could be checked in PDF attached.

Response: Special thanks to you! We have checked and corrected the entire manuscript according to your comments. Detailed revisions have been incorporated into the new version of the manuscript.

Round 2

Reviewer 2 Report

The authors have incorporated all suggestions and comments into the revised version, now the manuscript seems much clear. There is minor point to be corrected:

Ls.178 and 375: “de novo” should be in italic

Ls.196-199: Sentence should be in discussion section.

L.415: Delete “significantly”

Author Response

Comments and Suggestions for Authors

The authors have incorporated all suggestions and comments into the revised version, now the manuscript seems much clear. There is minor point to be corrected:

Response: Thank you again for your positive comments and valuable suggestions to improve the quality of our manuscript. We have examined the comments carefully and have made corrections to the manuscript accordingly, and the revised portions of the manuscript text are marked in blue. Our specific responses to the comments are listed as follows:

Ls.178 and 375: “de novo” should be in italic

Response: Thanks for your suggestion. We have checked and corrected the entire manuscript according to your comments.

Ls.196-199: Sentence should be in discussion section.

Response: According to your suggestion, we have moved the sentence (Lines 196-1991) to discussion. (Lines 388-390)

L.415: Delete “significantly”

Response: Thanks for your suggestion. We have deleted “significantly” in the new version of manuscript. (Line 415)